# Proposed Algorithm for Management of Meibomian Gland Dysfunction Based on Noninvasive Meibography

**DOI:** 10.3390/jcm10010065

**Published:** 2020-12-27

**Authors:** Reiko Arita, Shima Fukuoka, Motoko Kawashima

**Affiliations:** 1Department of Ophthalmology, Itoh Clinic, 626-11 Minami-Nakano, Minumaku, Saitama 337-0042, Japan; 2Lid and Meibomian Gland Working Group (LIME), Tokyo 112-0006, Japan; fshima3271@gmail.com (S.F.); motoko326@gmail.com (M.K.); 3Omiya Hamada Eye Clinic, 1-169-1, Sakuragicho, Omiyaku, Saitama 330-0854, Japan; 4Department of Ophthalmology, Keio University, 35 Shinanomachi, Shinjuku-ku, Tokyo 160-8582, Japan

**Keywords:** meibomian gland, meibography, meiboscore, meibomian gland dysfunction

## Abstract

Although the pathophysiology of meibomian gland dysfunction (MGD) remains incompletely understood, many treatment options have recently become available. According to an international workshop report, treatment selection for MGD should be based on a comprehensive stage classification dependent on ocular symptoms, lid margin abnormalities, meibum grade, and ocular surface staining. However, it is often difficult to evaluate all parameters required for such classification in routine clinical practice. We have now retrospectively evaluated therapeutic efficacy in MGD patients who received five types of treatment in the clinic setting: (1) meibocare (application of a warm compress and practice of lid hygiene), (2) meibum expression plus meibocare, (3) azithromycin eyedrops plus meibocare, (4) thermal pulsation therapy plus meibocare, or (5) intense pulsed light (IPL) therapy plus meibocare. Patients in each treatment group were classified into three subsets according to the meiboscore determined by noncontact meibography at baseline. Eyes in the IPL group showed improvement even if the meiboscore was high (5 or 6), whereas meibocare tended to be effective only if the meiboscore was low (1 or 2). The meiboscore may thus serve to guide selection of the most appropriate treatment in MGD patients. Prospective studies are warranted to confirm these outcomes.

## 1. Introduction

Meibomian gland dysfunction (MGD) is the leading cause of dry eye [1] and has a prevalence that varies widely from 3.5% to 70% according to age, sex, and ethnicity [2]. A population-based study (Hirado-Takushima study) performed on Takushima island in Japan found the prevalence of MGD to be 32.3% [3]. A survey of cataract surgery patients found that 63% of such individuals showed signs of MGD, and MGD was found to adversely affect visual acuity and patient satisfaction after such surgery [4]. According to the European Society of Cataract and Refractive Surgeons (ESCRS) and American Society of Cataract and Refractive Surgery (ASCRS) guidelines for cataract surgery, MGD should be diagnosed and treated before such surgery [5].

MGD is a chronic condition of the meibomian glands that is characterized by terminal duct obstruction or qualitative or quantitative changes in glandular secretion (meibum) [6]. In the obstructive form of MGD, hyperkeratinization of the ductal epithelium results in a reduced availability of meibum to coat the aqueous layer of the tear film [7]. This meibum deficiency thus gives rise to increased tear evaporation and consequent tear hyperosmolarity [8].

MGD is diagnosed on the basis of subjective symptoms, lid margin abnormalities, the condition of the gland orifices, and meibum grade [9]. Approaches such as conventional meibography and confocal microscopy for observation of the morphology of meibomian glands as well as tear interferometry for evaluation of gland function are also available [9], but they are not widely adopted in the clinic. Noncontact meibography is a recently developed noninvasive method that allows relatively rapid imaging of meibomian glands [10] with high reproducibility and which yields images convincing to patients of the need for treatment [11]. It is now widely adopted in clinical practice for evaluation of meibomian gland–related diseases.

Treatment options for MGD have increased greatly—in particular, with the recent advent of nonpharmaceutical treatments [12]—since the International Workshop on Meibomian Gland Dysfunction in 2011 [8]. Selection of a treatment for MGD is currently based on the stage classification proposed at the 2011 workshop [13]. Such stage classification is itself based on a comprehensive evaluation of subjective symptoms, lid margin abnormalities (plugging, vascularity), meibum grade, and degree of ocular surface staining [13]. However, it is often difficult to select a treatment method according to this complicated classification in the clinic. Moreover, it is unclear at what stage nonpharmaceutical treatment options, such as intraductal probing [14], thermal pulsation therapy [15], and intense pulsed light [16], should be performed.

We have therefore now conducted a retrospective examination of the characteristics of MGD patients who visited Itoh Clinic and received one of five types of treatment. The efficacy of each treatment was reevaluated from the viewpoint of noninvasive meibography grading (meiboscore) at baseline [10].

## 2. Experimental Section

This retrospective randomized study was conducted at Itoh Clinic in Saitama, Japan, adhered to the tenets of the Declaration of Helsinki, and was approved by the Institutional Review Board of the Faculty of Medicine at Itoh Clinic (approval code: IRIN201302-05). Written informed consent was obtained from all participants.

### 2.1. Patients and Treatment

Patients with MGD who attended Itoh Clinic between April 2014 and September 2020 were eligible for enrollment. One clinician (R.A.) who is an expert on MGD diagnosed the condition and enrolled MGD patients. The patients were consecutively enrolled in the study, with their baseline characteristics being found not to differ significantly among the treatment groups. Inclusion criteria were as follows: (1) an age of at least 20 years; (2) a diagnosis of MGD according to Japanese diagnostic criteria [17] including ocular symptoms, plugged gland orifices, vascularity and irregularity of lid margins, and decreased meibum quality and quantity (Shimazaki grading) [18]. Exclusion criteria comprised active ocular infection, ocular inflammatory disease, or aqueous-deficient dry eye (Schirmer test value of ≤5 mm). All enrolled patients performed meibocare, defined as warming of eyelids and the practice of lid hygiene twice a day. Five types of therapy were conducted during the study period: (1) meibocare alone for 3 months (years 2014–2016), (2) four sessions of meibum expression with an Arita meibomian gland compressor (Katena) 3 weeks apart together with meibocare over 3 months (MGX group) (years 2015–2016), (3) instillation of azithromycin eyedrops, Azimychin, Senju) for 2 weeks together with meibocare (AZM group) (years 2019–2020), (4) one session of treatment with a LipiFlow thermal pulsation system (Johnson & Johnson) together with meibocare for 1 month (years 2015–2017), and (5) four sessions of treatment with intense pulsed light (M22, Lumenis) 3 weeks apart together with meibocare over 3 months (IPL group) (years 2016–2019). All patients were allowed to apply artificial tears four times a day. All patients were examined before and1 month after the end of the treatment period.

### 2.2. Clinical Examinations

Ocular symptoms were assessed with the Standardized Patient Evaluation of Eye Dryness (SPEED) questionnaire [19]. The thickness of the lipid layer of the tear film (LLT) was measured with a LipiView interferometer (Johnson & Johnson). Lid margin abnormalities [20]—including plugging (scale of 0–3) and vascularity (scale of 0–3)—as well as the fluorescein-based breakup time of the tear film (FBUT), corneal-conjunctival fluorescein staining (fluo) score (scale of 0–9) [21], and grade of meibum expressed with digital pressure (scale of 0–3) [18] were evaluated by slitlamp microscopy. The meiboscore (0–3 for each eyelid), which reflects the extent of meibomian gland loss, was determined with a noncontact meibography system (Topcon) [10], and the meiboscore for both eyelids was summed (total of 0–6) (Figure 1) [10]. The volume of tear fluid was measured by Schirmer’s test performed without the administration of anesthetic [22]. Eyes were categorized as showing an improvement (that is, treatment was effective) if the SPEED score had decreased by ≥4 points [23] and meibum grade had decreased by ≥1 point after treatment compared with before treatment. Data for this study were obtained from the right eye of each subject unless the right eye was excluded, in which case data from the left eye were used.

### 2.3. Statistical Analysis

Data were found to be nonnormally distributed with the Shapiro-Wilk test (*p* < 0.05), and nonparametric testing was therefore applied. Baseline variables were compared among the treatment groups with Dunn’s multiple-comparison test. The Wilcoxon signed-rank test was used to compare variables between baseline and after treatment. The outcome variables of the study were the SPEED score and meibum grade before and after treatment. We performed a statistical power analysis for both the SPEED score and meibum grade. For the SPEED score, the mean difference between before after treatment was 5.3, with a corresponding SD of 4.5; for meibum grade, the mean difference was 1.1 with an SD of 1.0. These changes were calculated from the results of all 165 eyes in the current study. The average number of eyes in each group was 33. The power (1 − β) was 0.91 and 0.86 at the level of α = 0.05 for the SPEED score and meibum grade, respectively, and the sample size was sufficient. Statistical analysis was performed with JMP Pro version 15 software (SAS). Statistical tests were two sided, and a *p* value of <0.05 was considered statistically significant.

## 3. Results

### 3.1. Patient Characteristics

A total of 165 patients was enrolled in this study. The characteristics of the study subjects before and after treatment are presented in Table 1. None of the measured parameters at baseline differed significantly among the five treatment groups (Table 2).

Previous therapies for the enrolled patients are shown in Table 3, with most individuals having been prescribed eyedrops including hyaluronic acid eyedrops, preservative-free artificial tears, diquafosol eyedrops, topical steroid eyedrops, and rebamipide eyedrops. About half of the patients had performed meibocare or undergone meibomian gland expression. None of them had previously received azithromycin eyedrops, LipiFlow treatment, or intense pulsed light therapy.

### 3.2. Treatment Efficacy

The SPEED score was significantly reduced at 1 month after the end of the treatment period for all treatment groups (Table 1). LLT was significantly increased in the MGX group and the LipiFlow group. Plugging was significantly improved in all groups with the exception of the meibocare group. Vascularity was significantly improved in the AZM group, the LipiFlow group, and the IPL group. FBUT was significantly prolonged in all groups with the exception of the AZM group and the LipiFlow group. The fluo score was significantly decreased in all groups. Meibum grade was significantly improved in all groups.

### 3.3. Treatment Efficacy According to the Meiboscore

Eyes in each treatment group were graded on the basis of the meiboscore (1 or 2, mild gland loss; 3 or 4, moderate gland loss; 5 or 6, severe gland loss) at baseline (Figure 1). Eyes were also categorized as showing improvement (treatment was effective) if the SPEED score decreased by ≥4 points and meibum grade decreased by ≥1 point compared with the values before treatment. In the meibocare group, 100% of patients with mild gland loss showed improvement (Table 4). However, none of those with moderate or severe gland loss showed improvement. In the MGX group, 88% of patients with mild gland loss improved, compared with 55% of those with moderate gland loss and none of those with severe gland loss. In the AZM group, 92% of patients with mild gland loss and 100% of those with moderate gland loss improved, whereas none of those with severe gland loss did so. In the LipiFlow group, 75%, 10%, and 0% of patients with mild, moderate, or severe gland loss, respectively, showed improvement. Finally, in the IPL group, all patients with mild to severe gland loss improved.

## 4. Discussion

Diagnosis of MGD is largely made on the basis of the combination of subjective symptoms and the findings of slitlamp microscopy [9], but the guidelines for treatment selection according to disease severity are not clear. Given that more than half of MGD patients have no symptoms [20] and it is difficult to estimate the disease duration [20], existing guidelines are insufficient for accurate determination of MGD severity. We have now performed a retrospective assessment of the efficacy of five different types of treatment based on the meiboscore for MGD patients who attended Itoh Clinic. Our findings suggest that some treatment options can be selected according to the extent of disruption of meibomian gland morphology apparent in images obtained by noncontact meibography.

Meibography has been improved substantially since its introduction by Tapie in 1977 [24], but it was originally invasive and was not widely applied clinically. The development of noninvasive meibography based on infrared light [10] made it possible to observe meibomian glands of patients in general clinical practice, and it has served as a basis for many clinical studies [11,25]. It has thus not only revealed changes in meibomian gland morphology associated with various ocular surface diseases and provided insight into disease pathophysiology [25], but also highlighted the importance of diagnosis and treatment of MGD in many types of ophthalmology patients, including those treated with antiglaucoma eyedrops [26,27] or undergoing cataract surgery [4], as well as children and adolescents [28,29]. The specificity and sensitivity for MGD diagnosis based on the morphology of meibomian glands are high at 85% and 96.7%, respectively [30]. Meibography images convince patients of the need for treatment and are useful for obtaining informed consent in clinical studies. However, it is difficult to finely quantify meibomian gland area in such images, and they are not suitable for monitoring because gland area is not readily recovered by treatment. Tests of meibomian gland function such as meibum grading are relatively subjective. Tear interferometry is quantitative, but the findings are readily influenced by conditions such as humidity, room and body temperature, and eye makeup. On the other hand, meibography is objective and highly reproducible and can accurately diagnose MGD [31] and evaluate disease status [31,32]. Assessment of both gland morphology and function would be the ideal way to evaluate the efficacy of MGD treatment in the future. Given the retrospective nature of the present study, however, the efficacy of MGD treatment was evaluated by one expert clinician according to the meiboscore in order to minimize potential bias.

The five types of treatment performed at Itoh Clinic during the study period are administered (prescribed) for MGD in Japan. Given that the times the various treatments were launched in Japan differ, the times they were performed also differed. Patients were consecutively enrolled in the study, and there was no significant difference in baseline characteristics among the treatment groups. All five treatment types significantly improved subjective symptoms and objective tear parameters compared with baseline. However, our analysis of treatment efficacy according to the meiboscore at baseline revealed that meibocare tended to be effective only for eyes at the mild stage of MGD and that intense pulsed light was effective at all stages and was the only effective treatment for eyes at the severe stage of the disease characterized by many gland dropouts. The efficacies of meibomian gland expression and the LipiFlow device tended to be similar, consistent with the results of a previous study suggesting that LipiFlow is not effective in eyes with many gland dropouts [33]. We defined improvement of eyes with MGD as a decrease in the SPEED score of ≥4 points and a decrease in meibum grade of ≥1 point in our study. Our results do not imply that intense pulsed light induced regeneration of meibomian glands, however. The efficacy of intense pulsed light may depend on an anti-inflammatory action as well as on melting of meibum, and it may therefore be more effective for severe MGD associated with many gland dropouts than is the LipiFlow device, whose efficacy is thought to rely on meibum melting and gland massage. In addition, the standard protocol for intense pulsed light therapy applied in the present study consists of four sessions at 3-week intervals, whereas the standard protocol for LipiFlow is a single application. This difference might have affected the results of our study. Moreover, MGD patients treated with the LipiFlow device showed a significant improvement in symptoms and most signs.

In the present study, we summed the meiboscores for the upper and lower eyelids [10]. It remains controversial, however, whether the meiboscore should be evaluated for the upper eyelid alone, the lower eyelid alone, or both eyelids [34]. Changes apparent in the upper and lower eyelids are not always similar, and it is important to determine the reserve capacity of both upper and lower meibomian glands from the viewpoint of the oil reservoir for coating the entire ocular surface. With regard to assessment of the severity of MGD and selection of a treatment method, it would be desirable to make a comprehensive judgment based on the sum of the meiboscores of the upper and lower meibomian glands.

Tear film breakup time was measured with fluorescein in this study, which began before publication of the TFOS DEWS II report [35] that recommended the use of noninvasive measures of tear film stability. The invasiveness of the procedure in the present study was minimized by gently applying the fluorescein-stained paper to the conjunctival sac and asking the patient to blink twice.

Limitations of the present study include its retrospective nature, with the result that there was no washout period for previous treatments prior to the selected treatment. All patients also received meibocare in addition to the selected specific treatment option. In addition, the treatment periods and applications were set according to the treatment protocols and so differed among the patient groups. It should be considered in the future whether these differences can be minimized.

In conclusion, our results suggest that the most appropriate treatment can be selected for each MGD patient on the basis of the meiboscore. When meibomian gland loss is early and mild or moderate, several treatment options are available. When meibomian gland loss is severe, intense pulsed light treatment is recommended. Meibography may thus predict the effectiveness of future treatment and thereby inform selection of the best treatment option for each patient, especially for individuals with many gland dropouts. Future prospective studies are needed to confirm the outcomes of the present study.

## Figures and Tables

**Figure 1 jcm-10-00065-f001:**
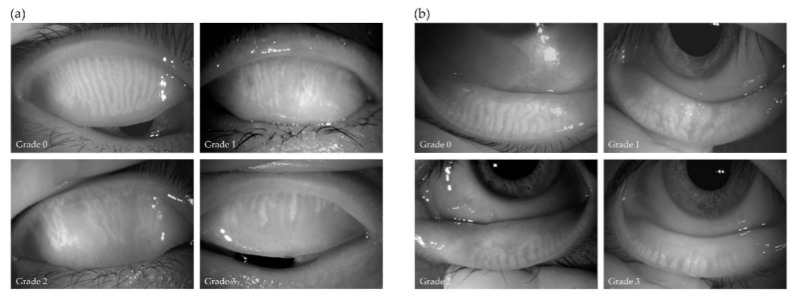
Representative images obtained by noncontact meibography from upper (**a**) and lower (**b**) eyelids with a meiboscore of 0 to 3. The lost area of meibomian glands was graded as 0 for no loss (upper left panels), 1 for a lost area of less than one-third (upper right panels), 2 for a lost area of between one-third and two-thirds (lower left panels), or 3 for a lost area of more than two-thirds (lower right panels).

**Table 1 jcm-10-00065-t001:** Characteristics of the study subjects with meibomian gland dysfunction before and after treatment in the meibocare, meibomian gland expression (MGX), azithromycin eyedrop (AZM), LipiFlow, and intense pulsed light (IPL) groups.

	Pre- or Posttreatment	Meibocare	MGX	AZM	LipiFlow	IPL
Characteristic		(*n* = 30 Eyes)	(*n* = 30 Eyes)	(*n* = 38 Eyes)	(*n* = 30 Eyes)	(*n* = 37 Eyes)
Sex (male/female)		17/13	16/14	20/18	16/14	22/15
Age (years)		59.1 ± 18.7	58.9 ± 15	60.2 ± 16	62.9 ± 14.2	60.5 ± 18
Duration of dry eye (years)		4.4 ± 2.5	4.2 ± 2.3	4.4 ± 2.2	4.3 ± 3.5	4.3 ± 2.3
SPEED score (0–28)	Pre	12.2 ± 3.9	12.5 ± 4.2	12.6 ± 4	11.3 ± 3	13.4 ± 3.2
	Post	9.0 ± 3.6	8.9 ± 4.9	6.2 ± 4.5	8.7 ± 4.2	3.7 ± 2.8
	*p* value	<0.001 **	<0.001 **	<0.001 **	<0.001 **	<0.001 **
LLT (nm)	Pre	54.8 ± 19.6	53.4 ± 11.6	57.9 ± 20.5	59.8 ± 17.9	56.2 ± 23.7
	Post	53.0 ± 18.3	59.3 ± 14.8	61.7 ± 19.2	65.2 ± 27.8	65.4 ± 22.5
	*p* value	0.11	0.031 *	0.36	0.015*	0.10
Plugging (0–3)	Pre	2.1 ± 0.8	2.1 ± 0.8	2.0 ± 0.6	2.0 ± 0.8	2.3 ± 0.9
	Post	1.9 ± 0.9	1.6 ± 0.9	0.9 ± 0.8	1.4 ± 0.9	0.2 ± 0.4
	*p* value	0.057	<0.001 **	<0.001 **	<0.001 **	<0.001 **
Vascularity (0–3)	Pre	1.3 ± 0.5	1.4 ± 0.9	1.6 ± 0.6	1.7 ± 1.0	1.4 ± 0.7
	Post	1.3 ± 0.4	1.4 ± 0.9	0.6 ± 0.6	1.4 ± 1.0	0.2 ± 0.4
	*p* value	0.33	1	0.006 *	<0.001 **	<0.001 **
FBUT (s)	Pre	3.1 ± 1.2	3.0 ± 1.1	3.2 ± 1.0	3.3 ± 0.7	3.1 ± 1.2
	Post	3.4 ± 1.3	3.9 ± 0.6	5.8 ± 2.8	3.1 ± 1.7	6.7 ± 2.4
	*p* value	0.030 *	<0.001 **	0.37	<0.001 **	<0.001 **
Fluo score (0–9)	Pre	0.8 ± 0.6	0.8 ± 0.8	1.0 ± 1.2	0.9 ± 0.6	1.0 ± 1.1
	Post	0.6 ± 0.6	0.5 ± 0.7	0.4 ± 1.0	0.8 ± 0.6	0.1 ± 0.3
	*p* value	0.006 *	0.001 **	0.043 *	<0.001 **	<0.001 **
Meibum grade (0–3)	Pre	2.3 ± 0.5	2.4 ± 0.5	2.3 ± 0.4	2.3 ± 0.4	2.6 ± 0.5
	Post	1.8 ± 0.9	1.6 ± 0.9	1.2 ± 0.8	1.7 ± 1.0	0.1 ± 0.4
	*p* value	<0.001 **	<0.001 **	<0.001 **	<0.001 **	<0.001 **
Meiboscore (0–6)	Pre	3.8 ± 1.7	3.8 ± 1.7	3.8 ± 1.6	3.8 ± 1.8	3.9 ± 1.5
Schirmer test value (mm)	Pre	10.9 ± 4.2	12.4 ± 9.1	10.6 ± 5.8	9.7 ± 5.6	11.0 ± 6.7
	Post	10.3 ± 4.3	12.0 ± 9	8.5 ± 4.6	9.0 ± 7.3	10.6 ± 6.9
	*p* value	0.056	0.169	0.16	0.11	0.003 *

Data are means ± SD. *p* values for comparisons between pre- and posttreatment were determined with the Wilcoxon signed-rank test (* *p* < 0.05, ** *p* < 0.001). SPEED, Standardized Patient Evaluation of Eye Dryness; LLT, lipid layer thickness of the tear film; FBUT, tear film breakup time with fluorescein; fluo, fluorescein staining.

**Table 2 jcm-10-00065-t002:** Comparison of the baseline characteristics of the study subjects with meibomian gland dysfunction in the meibocare, meibomian gland expression (MGX), azithromycin eyedrop (AZM), LipiFlow, and intense pulsed light (IPL) groups.

Characteristic		*p* Value			
Age	Treatment	Meibocare	MGX	AZM	LipiFlow
	MGX	1			
	AZM	1	1		
	LipiFlow	1	1	1	
	IPL	1	1	1	1
Duration of dry eye	Treatment	Meibocare	MGX	AZM	LipiFlow
	MGX	1			
	AZM	1	1		
	LipiFlow	1	1	1	
	IPL	1	1	1	1
SPEED score	Treatment	Meibocare	MGX	AZM	LipiFlow
	MGX	1			
	AZM	1	1		
	LipiFlow	1	1	1	
	IPL	0.62	1	1	0.098
LLT	Treatment	Meibocare	MGX	AZM	LipiFlow
	MGX	1			
	AZM	1	1		
	LipiFlow	1	1	1	
	IPL	1	1	1	1
Plugging	Treatment	Meibocare	MGX	AZM	LipiFlow
	MGX	1			
	AZM	1	1		
	LipiFlow	1	1	1	
	IPL	1	1	0.50	0.99
Vascularity	Treatment	Meibocare	MGX	AZM	LipiFlow
	MGX	1			
	AZM	0.68	1		
	LipiFlow	0.44	1	1	
	IPL	1	1	1	1
FBUT	Treatment	Meibocare	MGX	AZM	LipiFlow
	MGX	1			
	AZM	1	1		
	LipiFlow	1	1	1	
	IPL	1	1	1	1
Fluo score	Treatment	Meibocare	MGX	AZM	LipiFlow
	MGX	1			
	AZM	1	1		
	LipiFlow	1	1	1	
	IPL	1	1	1	1
Meibum grade	Treatment	Meibocare	MGX	AZM	LipiFlow
	MGX	1			
	AZM	1	1		
	LipiFlow	1	1	1	
	IPL	0.24	0.90	0.062	0.11
Meiboscore	Treatment	Meibocare	MGX	AZM	LipiFlow
	MGX	1			
	AZM	1	1		
	LipiFlow	1	1	1	
	IPL	1	1	1	1
Schirmer test value	Treatment	Meibocare	MGX	AZM	LipiFlow
	MGX	1			
	AZM	1	1		
	LipiFlow	0.83	1	1	
	IPL	1	1	1	1

*p* values were determined with Dunn’s multiple-comparison test. SPEED, Standardized Patient Evaluation of Eye Dryness; LLT, lipid layer thickness of the tear film; FBUT, tear film breakup time with fluorescein; fluo, fluorescein staining.

**Table 3 jcm-10-00065-t003:** Previous therapies for the study patients with meibomian gland dysfunction in the meibocare, meibomian gland expression (MGX), azithromycin eyedrop (AZM), LipiFlow, and intense pulsed light (IPL) groups.

	No. (%) of Patients				
Therapy	Meibocare (*n* = 30)	MGX (*n* = 30)	AZM (*n* = 38)	LipiFlow (*n* = 30)	IPL (*n* = 37)
Hyaluronic acid eyedrops	17 (56.7%)	20 (66.7%)	25 (65.8%)	25 (83.3%)	30 (81.1%)
Preservative-free artificial tears	15 (50.0%)	18 (60.0%)	15 (39.5%)	10 (33.3%)	14 (37.8%)
Diquafosol eyedrops	5 (16.7%)	7 (23.3%)	10 (26.3%)	15 (50.0%)	20 (54.1%)
Topical steroid eyedrops	8 (26.7%)	6 (20.0%)	10 (26.3%)	9 (30.0%)	10 (27.0%)
Rebamipide eyedrops	3 (10.0%)	5 (16.7%)	6 (15.8%)	8 (26.7%)	10 (27.0%)
Meibocare	-	12 (40.0%)	16 (42.1%)	15 (50.0%)	20 (54.1%)
Meibomian gland expression	12 (40.0%)	-	15 (39.4%)	15 (50.0%)	15 (40.5%)
Azithromycin eyedrops	0 (0.0%)	0 (0.0%)	-	0 (0.0%)	0 (0.0%)
LipiFlow	0 (0.0%)	0 (0.0%)	0 (0.0%)	-	0 (0.0%)
Intense pulsed light	0 (0.0%)	0 (0.0%)	0 (0.0%)	0 (0.0%)	-

**Table 4 jcm-10-00065-t004:** Improvement of eyes with meibomian gland dysfunction after treatment in the meibocare, meibomian gland expression (MGX), azithromycin eyedrop (AZM), LipiFlow, and intense pulsed light (IPL) groups according to the meiboscore at baseline.

	Meibocare	MGX	AZM	LipiFlow	IPL
Meiboscore	Improved/Total	Improved/Total	Improved/Total	Improved/Total	Improved/Total
1 or 2	8/8 (100%)	7/8 (88%)	11/12 (92%)	6/8 (75%)	10/10 (100%)
3 or 4	0/11 (0%)	6/11 (55%)	13/13 (100%)	1/10 (10%)	14/14 (100%)
5 or 6	0/11 (0%)	0/11 (0%)	0/13 (0%)	0/12 (0%)	13/13 (100%)

Eyes were categorized as showing improvement if the SPEED score decreased by ≥4 points and meibum grade decreased by ≥1 point compared with the values before treatment.

## Data Availability

The data presented in this study are available on request from the corresponding author.

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
