# Peer review of "Proposed Algorithm for Management of Meibomian Gland Dysfunction Based on Noninvasive Meibography"

_jcm, 2020, doi:10.3390/jcm10010065_

Round 1
Reviewer 1 Report
Authors performed an interesting study addressing the efficacy of five common clinical treatments for MGD patients. However, data was retrospectively recorded and thus, the study design has inherent limitations.
Methods.
Authors must state how many clinicians performed the clinical tests, specially the ones that requires subjective grading (i.e. Meiboscore, lid margin abnormalities, etc). Authors indicated that MGD patients attended Itoh Clinic, it is likely that patient were diagnosed by various clinicians. And it is also well-known that subjective test scores depend on the inter-observer variability.
Authors criteria for improvement included the following “… the SPEED score had decreased by ≥3 points…”. Why was selected this cut off value? Is there any study determining the minimal clinically important difference (MCID) for SPEED like OSDI (Arch Ophthalmol. 2010;128:94-101)? It is important to know if statistically significant variations exists after treatment, but it is even more important to know if these variations are clinically meaningful from patients viewpoint (questionnaire outcomes).
Line 113. Authors stated that they performed a statistical power analysis for both the SPEED score and meibum grade after the final treatment session in the control and IPL-MGX groups and concluded that the sample size was sufficient. Please provide further details so that it can be checked that n=30 for each group is sufficient.
Results
In table 1, it seems that most of the pre-treatment test values for the 5 groups are quite similar, however, it must be statistically confirmed by authors. This information is important to know if the patients recruited in each group differed before treatment.
I recommend authors to include the p-values described in table 3 in the table 1, thus, readers will be able to see at the same time the variation in each as well as if this variation is statistically significant.
Conclussions
Due to the retrospective nature of the study, it has numerous limitations already stated by authors. Nonetheless, manuscript text and abstract conclusions must indicate that future prospective studies are needed to confirm these outcomes.
Author Response
Responses to Reviewers
We thank the reviewers for the helpful comments on our manuscript, our specific responses to which are as follows.
Response to Reviewer 1
Authors performed an interesting study addressing the efficacy of five common clinical treatments for MGD patients. However, data was retrospectively recorded and thus, the study design has inherent limitations.
- Methods.
Authors must state how many clinicians performed the clinical tests, specially the ones that requires subjective grading (i.e. Meiboscore, lid margin abnormalities, etc). Authors indicated that MGD patients attended Itoh Clinic, it is likely that patient were diagnosed by various clinicians. And it is also well-known that subjective test scores depend on the inter-observer variability.
RESPONSE: One clinician (R.A.) who is an expert on MGD diagnosed the condition. We have now added this information to the methods section (page 2, lines 77–78).
- Authors criteria for improvement included the following “… the SPEED score had decreased by ≥3 points…”. Why was selected this cut off value? Is there any study determining the minimal clinically important difference (MCID) for SPEED like OSDI (Arch Ophthalmol. 2010;128:94-101)? It is important to know if statistically significant variations exists after treatment, but it is even more important to know if these variations are clinically meaningful from patients viewpoint (questionnaire outcomes).
RESPONSE: We now cite a previous study (Asiedu et al. Cornea 2016; 35: 175–180) in this regard and have reanalyzed our data with the SPEED score cutoff at ≥4 points. Eyes were thus categorized as showing an improvement (that is, treatment was effective) if the SPEED score had decreased by ³4 points and meibum grade had decreased by ³1 point after treatment compared with before treatment. We changed the methods (page 3, line 109) and results (page 9, line 191) sections as well as Table 4 accordingly.
- Line 113. Authors stated that they performed a statistical power analysis for both the SPEED score and meibum grade after the final treatment session in the control and IPL-MGX groups and concluded that the sample size was sufficient. Please provide further details so that it can be checked that n=30 for each group is sufficient.
RESPONSE: We have now provided further details of the power calculation in the methods section (page 4, lines 129–133).
- Results
In table 1, it seems that most of the pre-treatment test values for the 5 groups are quite similar, however, it must be statistically confirmed by authors. This information is important to know if the patients recruited in each group differed before treatment.
RESPONSE: We show that the baseline characteristics did not differ significantly among the treatment groups in Table 2.
- I recommend authors to include the p-values described in table 3 in the table 1, thus, readers will be able to see at the same time the variation in each as well as if this variation is statistically significant.
RESPONSE: As suggested, we have now moved the P values from the original Table 3 to Table 1.
- Conclussions
Due to the retrospective nature of the study, it has numerous limitations already stated by authors. Nonetheless, manuscript text and abstract conclusions must indicate that future prospective studies are needed to confirm these outcomes.
RESPONSE: As requested, we have now pointed out the need for future prospective studies at the end of the abstract (page 1, lines 30-31) and discussion (page 12, lines 293-294) sections.
Reviewer 2 Report
This retrospective study provides useful guidance for clinicians in predicting the likely success of various treatments based on a patient’s baseline MGD status. There are some limitations related to the retrospective nature of the study, variations in the length of treatment course between therapies, and possible confounding factors relating to prior treatment, but in the absence of high quality prospective RCTs, this study presents novel and helpful information.
- How was even and unbiased treatment allocation ensured (and managed) across the breadth of disease severity when this was a retrospective study conducted over 6 years? One might expect biases to exist in the selection of treatments for the patients according to expert clinician opinion? e.g. severe patients may be less likely to be offered meibocare only. Patients with telangiectasia/vascularity might be more likely offered IPL. Please comment as to how treatments were chosen for individual patients and what influence patients had in the choice of their treatment.
- The authors describe the lack of washout as a limitation of the study. What previous treatments were patients exposed to? How long had patients had dry eye and how many previous treatments had been applied?Were those in AZM, LipiFlow and IPL groups more likely to have had meibocare recommendations for many months/years before consideration of in office therapy and those in the meibocare group more likely to be patients with a more recent diagnosis of dry eye? Was there any risk of patients in the meibocare group having had AZM, LipiFlow or IPL treatment prior? The extent and implications of these limitations should be described more clearly.
- It’s not clear if all treatments were available across the full duration of the study (since 2014). Was IPL introduced to the clinic by 2014? Any differences in availability that could have introduced bias and inconsistencies in outcomes should be acknowledged.
- Meibography is a test that highlights MG morphology and as such indicates the capacity of the system, but not function.Given Korb reports that all glands are not functional simultaneously (around 45% only) (and ergo that all glands need not be present, to have a functional lipid layer), would it not be valuable to assess both structure and function in assessing baseline status since tear film homeostasis may not be compromised if there are a number of fully functional glands, even if some are missing on meibography. Sensitivity and specificity of meibography in identifying MGD is relatively high, but what additional benefit might have been obtained by combining meibography with lipid function via either expressibility (meibomian glands yielding lipid secretion) or tear film lipid layer quality?
- Fluorescein breakup time was used to describe the tear film stability. This study began before the publication of TFOS DEWS II and other literature which has recommended the use of non-invasive measures of tear film stability, but this may be deserving of a comment to justify use of this more invasive test, and any steps taken to minimize its invasiveness in the discussion section.
- Table 1 – it would be helpful to indicate significant differences between pre and post-treatment values with asterisks.
- Table 2 – Reconsider if this is the most helpful way to represent the data.It seems strange that this is reportedly describing comparison of baseline characteristics, yet IPL (which shows a similar mean and standard deviation at baseline) is flagged as being potentially different from the other groups. Is this actually post-treatment results that are shown in the table? This would make more sense according to the results shown in Table 1.
- Page 6 of 9; line 149-150: avoid using the word ‘change’ to describe the meibomian gland status at baseline as this is confusing and may be misconstrued as an improvement post-treatment.A possible alternative term might be ‘mild gland loss’. Similarly in Figure 1, suggest rewording: ‘lost area of meibomian glands’ and replacing with ‘area of meibomian gland loss’, throughout legend.
- Can the authors justify the significance of a change of ≥3 points in SPEED score or ≥1 decrease in meibum grade?This measure of success should be supported by statistical and clinical justification and/or from evidence in the literature.
- Figure 2 does not appear to add any information that is not described in Table 4. Retention of this figure may therefore need to be justified.
- How do the authors explain such treatment success with IPL in severely compromised meibomian gland morphology?What is the proposed mechanism of action of this treatment, that is so different from LipiFlow, that it should work 100% effectively in patients without many meibomian glands (i.e. with minimum reserve capacity) where LipiFlow was successful in 0%? Given this is the only treatment investigated which is successful with severely affected glands, and particularly as the success rate is 100% in this group, this finding needs to be discussed in much more detail.
- The conclusion does not seem entirely justified by the findings of the study.The results provide an estimate of likelihood for success with each individual therapy for a specified baseline meiboscore, but it does not predict which therapy is best for a specific grade, except for grades 5-6 which appear to improve only with IPL. For mild meibomian gland loss it seems that any of the treatments tested will be effective (there is no report of whether one is more effective than others based on the dichotomous definition of success/non-success). The conclusion should therefore be reworded to more accurately reflect the reported outcomes.
Author Response
Responses to Reviewers
We thank the reviewers for the helpful comments on our manuscript, our specific responses to which are as follows.
Response to Reviewer 2
This retrospective study provides useful guidance for clinicians in predicting the likely success of various treatments based on a patient’s baseline MGD status. There are some limitations related to the retrospective nature of the study, variations in the length of treatment course between therapies, and possible confounding factors relating to prior treatment, but in the absence of high quality prospective RCTs, this study presents novel and helpful information.
- How was even and unbiased treatment allocation ensured (and managed) across the breadth of disease severity when this was a retrospective study conducted over 6 years? One might expect biases to exist in the selection of treatments for the patients according to expert clinician opinion? e.g. severe patients may be less likely to be offered meibocare only. Patients with telangiectasia/vascularity might be more likely offered IPL. Please comment as to how treatments were chosen for individual patients and what influence patients had in the choice of their treatment.
RESPONSE: Given that the times the various treatments were launched in Japan differ, the times they were performed also differed. Patients were consecutively enrolled in the study, and there was no significant difference in baseline characteristics among the treatment groups. We have now clarified this point in the methods (page 2, lines 78-80, 87-93) and discussion (page 11, lines 249–252) sections.
- The authors describe the lack of washout as a limitation of the study. What previous treatments were patients exposed to? How long had patients had dry eye and how many previous treatments had been applied?Were those in AZM, LipiFlow and IPL groups more likely to have had meibocare recommendations for many months/years before consideration of in office therapy and those in the meibocare group more likely to be patients with a more recent diagnosis of dry eye? Was there any risk of patients in the meibocare group having had AZM, LipiFlow or IPL treatment prior? The extent and implications of these limitations should be described more clearly.
RESPONSE: We have now included a new Table 3 showing medical history related to previous therapies for MGD. We also added the duration of dry eye to Tables 1 and 2. In addition, we have addressed this issue in the results section (page 7, lines 159–163)
- It’s not clear if all treatments were available across the full duration of the study (since 2014). Was IPL introduced to the clinic by 2014? Any differences in availability that could have introduced bias and inconsistencies in outcomes should be acknowledged.
RESPONSE: We have now added the time period that each treatment was applied to the methods section (page 3, lines 87–93) as well as addressed this point in the discussion section (page 11, lines 249–252).
- Meibography is a test that highlights MG morphology and as such indicates the capacity of the system, but not function.Given Korb reports that all glands are not functional simultaneously (around 45% only) (and ergo that all glands need not be present, to have a functional lipid layer), would it not be valuable to assess both structure and function in assessing baseline status since tear film homeostasis may not be compromised if there are a number of fully functional glands, even if some are missing on meibography. Sensitivity and specificity of meibography in identifying MGD is relatively high, but what additional benefit might have been obtained by combining meibography with lipid function via either expressibility (meibomian glands yielding lipid secretion) or tear film lipid layer quality?
RESPONSE: Tests of meibomian gland function such as meibum grading tend to be relatively subjective. Tear interferometry is quantitative, but the findings are easily influenced by conditions such as humidity, room and body temperature, and eye makeup. On the other hand, meibography is objective and highly reproducible. As the reviewer points out, evaluation of both morphology and function would be the ideal way to assess the efficacy of MGD treatment in the future. Given the retrospective nature of our study, however, the efficacy of MGD treatment was evaluated by one expert clinician on the basis of the meiboscore in order to minimize possible bias. We have now addressed this issue in the discussion section (pages 10-11, lines 240–247).
- Fluorescein breakup time was used to describe the tear film stability. This study began before the publication of TFOS DEWS II and other literature which has recommended the use of non-invasive measures of tear film stability, but this may be deserving of a comment to justify use of this more invasive test, and any steps taken to minimize its invasiveness in the discussion section.
RESPONSE: As suggested, we have now dealt with this point in the discussion section (page 11, lines 278–281).
- Table 1 – it would be helpful to indicate significant differences between pre and post-treatment values with asterisks.
RESPONSE: As suggested, we have now moved the P values from the original Table 3 to Table 1.
- Table 2 – Reconsider if this is the most helpful way to represent the data.It seems strange that this is reportedly describing comparison of baseline characteristics, yet IPL (which shows a similar mean and standard deviation at baseline) is flagged as being potentially different from the other groups. Is this actually post-treatment results that are shown in the table? This would make more sense according to the results shown in Table 1.
RESPONSE: We confirmed that the P values shown in Table 2 are indeed for comparisons of pretreatment values of the measured parameters.
- Page 6 of 9; line 149-150: avoid using the word ‘change’ to describe the meibomian gland status at baseline as this is confusing and may be misconstrued as an improvement post-treatment.A possible alternative term might be ‘mild gland loss’. Similarly in Figure 1, suggest rewording: ‘lost area of meibomian glands’ and replacing with ‘area of meibomian gland loss’, throughout legend.
RESPONSE: As suggested, we changed the phrasing to refer to gland loss (page 9, lines 188–202). We also changed “lost area of meibomian glands” to “area of meibomian gland loss” in the legend to Figure 1.
- Can the authors justify the significance of a change of ≥3 points in SPEED score or ≥1 decrease in meibum grade?This measure of success should be supported by statistical and clinical justification and/or from evidence in the literature.
RESPONSE: We now cite a previous study (Asiedu et al. Cornea 2016; 35: 175–180) in this regard and have reanalyzed our data with the SPEED score cutoff at ≥4 points. Eyes were thus categorized as showing an improvement (that is, treatment was effective) if the SPEED score had decreased by ³4 points and meibum grade had decreased by ³1 point after treatment compared with before treatment. We changed the methods (page 3, line 109) and results (page 9, line 191) sections as well as Table 4 accordingly.
- Figure 2 does not appear to add any information that is not described in Table 4. Retention of this figure may therefore need to be justified.
RESPONSE: We deleted Figure 2.
- How do the authors explain such treatment success with IPL in severely compromised meibomian gland morphology?What is the proposed mechanism of action of this treatment, that is so different from LipiFlow, that it should work 100% effectively in patients without many meibomian glands (i.e. with minimum reserve capacity) where LipiFlow was successful in 0%? Given this is the only treatment investigated which is successful with severely affected glands, and particularly as the success rate is 100% in this group, this finding needs to be discussed in much more detail.
RESPONSE: We defined improvement of eyes with MGD as a decrease in the SPEED score of ³4 points and a decrease in meibum grade of ³1 point in our study. Our results do not imply that intense pulsed light induced regeneration of meibomian glands, however. The efficacy of intense pulsed light may depend on an anti-inflammatory action as well as on melting of meibum, and it may therefore be more effective for severe MGD associated with many gland dropouts than is the LipiFlow device, whose efficacy is thought to rely on meibum melting and gland massage. In addition, the standard protocol for intense pulsed light therapy applied in the present study consists of four sessions at 3-week intervals, whereas the standard protocol for LipiFlow is a single application. This difference might have affected the results of our study. Moreover, MGD patients treated with the LipiFlow device showed a significant improvement in symptoms and most signs. We have now addressed this issue in the discussion section (page 11, lines 259–269).
- The conclusion does not seem entirely justified by the findings of the study.The results provide an estimate of likelihood for success with each individual therapy for a specified baseline meiboscore, but it does not predict which therapy is best for a specific grade, except for grades 5-6 which appear to improve only with IPL. For mild meibomian gland loss it seems that any of the treatments tested will be effective (there is no report of whether one is more effective than others based on the dichotomous definition of success/non-success). The conclusion should therefore be reworded to more accurately reflect the reported outcomes.
RESPONSE: We have now modified the conclusion paragraph of the discussion section (page 12, lines 288–294) as follows: “When meibomian gland loss is early and mild or moderate, several treatment options are available. When meibomian gland loss is severe, intense pulsed light treatment is recommended. Meibography may thus predict the effectiveness of future treatment and thereby inform selection of the best treatment option for each patient, especially for individuals with many gland dropouts. Future prospective studies are needed to confirm the outcomes of the present study.”
Round 2
Reviewer 1 Report
The issues raised during the previous revision by this reviewer have been addressed.